# A Low-Protein Diet with a Renal-Specific Oral Nutrition Supplement Helps Maintain Nutritional Status in Patients with Advanced Chronic Kidney Disease

**DOI:** 10.3390/jpm11121360

**Published:** 2021-12-14

**Authors:** Owen J. Kelly, Meng-Chuan Huang, Hsin-Yin Liao, Chih-Ching Lin, Tsui-Yin Tung, Rhoda Wen-Yi Cheng, Michael Yao-Hsien Wang, Menaka Yalawar, Shang-Jyh Hwang

**Affiliations:** 1Abbott Nutrition, Global Scientific & Medical Affairs, Columbus, OH 43219, USA; drojkelly@gmail.com; 2Department of Nutrition and Dietetics, Kaohsiung Medical University Hospital, Kaohsiung 80756, Taiwan; mechhu@kmu.edu.tw (M.-C.H.); hsin770304@gmail.com (H.-Y.L.); 3School of Medicine, College of Medicine, Kaohsiung Medical University, Kaohsiung 80708, Taiwan; 4Division of Nephrology, Department of Medicine, Taipei Veterans General Hospital, Taipei 40705, Taiwan; lincc2@vghtpe.gov.tw; 5Department of Nutrition and Dietetics, Taipei Veterans General Hospital, Taipei 40705, Taiwan; tytung4@gmail.com; 6Medical Affairs, Abbott Nutrition, Taipei 104483, Taiwan; rhoda.cheng@abbott.com (R.W.-Y.C.); michael.wang1@abbott.com (M.Y.-H.W.); 7Biostatistician, Statistical Services, Cognizant Technologies Solution Pvt. Ltd., Bangalore 560092, India; menaka.shekarappa@abbott.com; 8Division of Nephrology, Department of Medicine, Kaohsiung Medical University Hospital, Kaohsiung 807377, Taiwan

**Keywords:** serum albumin, renal insufficiency, chronic, nutritional status, dietary supplements, diet, protein-restricted

## Abstract

A low-protein diet (LPD) is recommended to patients with non-dialysis advanced chronic kidney disease (CKD) for delaying renal function decline. However, this approach potentially prevents an adequate calorie and micronutrient intake. We examined the influence of an LPD including a renal-specific oral nutrition supplement (RONS) on the nutrition status of patients with stage 3b–5 CKD. This multicenter, open-label study prospectively enrolled patients over 18 years of age, with an estimated glomerular filtration rate (eGFR) between 10 and 45 mL/min/1.73 m^2^, serum albumin ≥3.0 g/dL, and body mass index ≤30 kg/m^2^. All participants implemented the LPD with one serving of RONS daily for 6 months. Daily energy intake, nutrition status, renal function, and quality of life were assessed before and after the intervention. Of 53 enrolled patients, 35 (66.0%) completed the study. We found that RONS use increased patients’ energy intake and maintained their serum albumin, nutritional status, and quality of life. Body weight and handgrip strength increased significantly at 6 months after enrollment (*p* = 0.0357); eGFR slightly decreased at 3 and 6 months after enrollment, suggesting that patients’ residual renal function was preserved. Our findings support the conclusion that patients with non-dialysis advanced CKD may benefit from additional RONS besides their regular diet. Patients with advanced CKD receiving RONS might achieve better nutrition and delay renal function decline.

## 1. Introduction

Chronic kidney disease is associated with high morbidity and mortality [1]. Its global prevalence is estimated at around 13.4%. In Taiwan, the prevalence of CKD lies similarly at 12%, and about 7% of the entire population suffers CKD from stage 3 to 5 [2]; however, only 4% of the patients with CKD are aware of this disease, and early-stage CKD is poorly recognized in Taiwan [2,3]. Promotion of CKD’s disease awareness and delaying renal function decline require a multidisciplinary approach, including lifestyle modification, nutrition intervention, medications, and allocating additional healthcare resources. The National Health Insurance Administration in Taiwan initiated a comprehensive CKD shared-care program for those with CKD at stages 3b–5 to slow down disease progression to end-stage renal disease (ESRD). The Taiwan Society of Nephrology has also established a committee consisting of nephrologists, nurses, and dieticians to enhance patients’ CKD education and staff training [4]. Multiple options are available for reducing disease and socioeconomic burdens among patients with CKD. Among these therapeutic options, dietary modification can be relatively convenient to decrease patients’ susceptibility to adverse influences.

In addition to restricting dietary phosphorus, potassium, and fluid intake, the mainstay of dietary recommendations for the non-dialyzed CKD population includes protein limitation and the optimization of energy intake [5]. A Cochrane review concluded that a very-low-protein diet could reduce the risk of renal progression to dialysis among patients with advanced non-diabetic CKD, since a higher dietary protein intake might be associated with glomerular hyperfiltration and accelerate glomerular damage [6]. Protein restriction also reduces nitrogenous waste production and minimizes uremia [7]. On the other hand, a higher energy intake is required, since patients with early-stage CKD are found to have an increased catabolism related to inflammation, leading to the development of protein–energy wasting (PEW) [8]. Adequate caloric intake can be an important concurrent treatment in addition to lowering dietary protein. However, LPD for patients with non-dialysis CKD may offer insufficient calories and induce malnutrition, further worsening PEW [6]. A recent meta-analysis found that the PEW increased the severity of CKD [9]. Another study further concluded that oral nutrition supplements might prevent nutritional status deterioration in patients receiving chronic dialysis [10]; however, little is known about the influence of oral nutritional supplementation in patients with earlier stages of CKD or in those with non-dialysis CKD. Huang et al. found that excess protein or inadequate calorie intake occurred with a lower estimated glomerular filtration rate (eGFR) in Taiwanese patients with CKD at stages 3 to 5 [11,12]. The need for an increased energy intake poses a dilemma for patients with advanced CKD who are receiving LPD. According to a U.S. National Health and Nutrition Examination Survey (NHANES), the mean protein intake of the general population was higher than the recommended level of 0.8 g/kg actual body weight per day. A more recent analysis of the NHANES found that older individuals actually had a lower daily protein intake (<0.8 g/kg/day), likely due to an overall reduction in nutrient and energy intake associated with ageing [13]. Similarly, Moore et al. discovered a reduction in protein intake with a greater CKD severity in the U.S. population [14]. These findings support the utility of LPD in managing patients with CKD, while maintaining sufficient energy intake potentially further benefits patients with advanced CKD. Therefore, it can be tempting to use a renal-specific oral nutrition supplement (RONS) as a convenient and effective approach to improve the nutritional status of patients with advanced CKD.

This study aimed to determine whether a therapeutic strategy consisting of providing LPD with an additional serving of RONS per day, along with nutrition counseling performed by registered dieticians, could maintain nutritional status in patients with non-dialysis stage 3b–5 CKD.

## 2. Materials and Methods

### 2.1. Participants and Study Design

This prospective, multicenter, single-arm, and open-label study (NCT02046746) was performed in accordance with the Declaration of Helsinki and approved by the Institutional Review Board of Kaohsiung Medical University Chung-Ho Memorial Hospital (approval number: KMUHIRB-2013-09-04(II)) and by that of Taipei Veterans General Hospital (approval number: VGHIRB 2014-10-003C). Written informed consent was obtained from all participants. These patients were enlisted in the Multidisciplinary Pre-ESRD Educational Program (MPE) established by the National Health Insurance in Taiwan. MPE is a program widely established in over 600 healthcare institutions in Taiwan. Patients enrolled in the MPE were educated and routinely followed up for clinical interviews, laboratory examinations, nutritional assessments, and evaluation of the feasibility of relevant interventions, in order to prevent CKD progression and the development of CKD-related complications.

Patients were eligible if their age was over 18 years at the screening visit, provided an informed consent, were male or nonpregnant female, had an eGFR between 10 and 45 mL/min/1.73 m^2^, a serum albumin level ≥3.0 g/dL, a body mass index (BMI) ≤30 kg/m^2^, HbA1c ≤9.0% at screening, were willing to follow the protocol, and were not expected to receive dialysis during the next 18 months. Exclusion criteria included those with type 1 diabetes mellitus, who presented a malnourished state defined as having a subjective global assessment (SGA) score between 1 and 3, had known infectious diseases, prior cardiovascular events, bleeding disorders, gastrointestinal illnesses, overt diabetic retinopathy or neuropathy, or significant neurological or psychiatric disorder, had been hospitalized within 3 months prior to enrollment; had elective surgery planned over the course of the study; had substances abuse including alcohol; consumed substances that would interfere with the study measures; were allergic to any study product ingredients.

### 2.2. Sample Size Calculation

The sample size was estimated using albumin data based on findings from the nutritional supplementation group in Montes-Delgado et al.’s study [15], using SAS Release 9.3. A sample size of 18 participants was obtained. Such number of subjects had 80% power to detect a difference in mean albumin (−0.661 mg/dL) from baseline to the 6th month based on a standard deviation (SD) of differences (±1.071 mg/dL). The calculation was performed by paired t-test with one-sided significance level of *p* < 0.05.

### 2.3. Nutrition Intervention

All participants received standard care provided by multi-disciplinary teams at the participating institutes. During the 6-month period of the intervention, all participants were counseled by registered dieticians at baseline, 3rd month, and 6th month. The dieticians assisted all participants in maintaining an LPD in conjunction with one serving of RONS (Abbott Suplena^®^/Nepro LP^®^, Abbott Nutrition, Taipei, Taiwan: 425 kcal, 11 g protein, 23 g fat, 46 g carbohydrate, vitamins, and minerals). Dietary principles for the LPD included avoiding foods rich in sodium, phosphorus, and potassium. The individualized conditions of each participant were taken into consideration to establish daily energy and protein intake targets. Portion control of food groups was individualized for each participant to meet protein and energy intake targets. Anthropometric measurements, laboratory tests, dietary evaluation, appetite assessment, and handgrip strength were performed at baseline, 3rd month, and 6th month after enrollment. Quality of life (QoL) was assessed at baseline and at the 6th month. To assess participants’ adherence to RONS consumption, they were requested to consume ≥75% of the prescribed servings.

### 2.4. Anthropometric Measurements

Body composition parameters including total body protein, skeletal muscle mass, fat-free mass, total body fat mass, and body fat were analyzed by bioelectrical impedance (Inbody 220, Seoul, Korea). Handgrip strength was measured with a hand-held dynamometer. The placement of the dynamometer was standardized by adjusting the distal interphalangeal joints of the finger to just below the handle, with the arm adducted and the elbow at flexed at 90°. The final strength was defined as the average of three measurements.

### 2.5. Laboratory Tests

A fasting blood sample was drawn from a superficial arm vein and analyzed for the following items: albumin, total cholesterol, triglycerides, low-density lipoprotein (LDL)–cholesterol, serum glucose, HbA1c, blood urea nitrogen (BUN), uric acid, creatinine, sodium, potassium, phosphorous, calcium, high-sensitivity C-reactive protein (CRP), homeostatic model assessment for insulin resistance (HOMA-IR, calculated as fasting glucose [mg/dL] × fasting insulin [µU/mL]/405). All biochemical measurements were analyzed by the Department of Medical Technology at the Kaohsiung Medical University Chung-Ho Memorial Hospital.

### 2.6. Nutrition-Related Parameters

Prior to each hospital clinic visit, the participants were instructed by the dieticians to record in a food diary all foods and beverages consumed in three days, including two weekdays and one weekend day, for nutrient intake assessment. Using the food diaries, the hospital dieticians evaluated the average intakes of total energy (kcal/day, kcal/kg/day) and protein (g/day, g/kg/day). All participants were scheduled to meet with the dietician at baseline, 3rd month, and 6th month to strengthen and encourage their adherence to the study protocol. Compliance to dietary energy intake (DEI) and dietary protein intake (DPI) was assessed and recorded by the dieticians.

### 2.7. Quality of Life

QoL was evaluated using World Health Organization Quality of Life-BREF (WHOQOL-BREF Taiwan Version) [16]. The WHOQOL-BREF instrument comprises 26 items, which measure the following broad domains: physical health, psychological health, social relationships, and environment.

### 2.8. Statistical Analysis

Both evaluable and intention-to-treat (ITT, defined as participants who consumed at least one study product per day) analyses were performed. Continuous variables were analyzed using parametric analyses unless the variable significantly deviated from the normal distribution, in which case suitable nonparametric analyses were used. The residuals from the parametric were used to check for deviation from the normal distribution. A variable was declared non-normal if the Shapiro–Wilk test *p*-value was <0.001. Categorical variables were analyzed using tests of association. The primary outcome of interest was changes in serum biochemistry, nutritional status, and quality of life from baseline to the 3rd and the 6th month, and these results were compared between different time points using Student’s t-test. All data are presented as mean ± SD unless otherwise indicated. Statistical analyses were performed using SAS version 9.4. The analysis of covariance model was fitted for the variables’ change from baseline to the 3rd month and from baseline to the 6th month for comparison between the non-diabetic and the diabetic groups with baseline as a covariate. For changes from the 3rd month to the 6th month, the analysis of variance model was fitted. When the data were not normal, a two-sample two-sided Wilcoxon rank-sum test was used to compare the groups.

## 3. Results

There were 53 eligible participants initially recruited. Of these, eight participants never consumed the RONS within the one-month period. During the 6-month intervention, 10 participants consumed less than 75% of RONS and were excluded. A total of 35 participants completed the study and all assessments. Figure 1 illustrates the tracking information based on extensions to the Consolidated Standards of Reporting Trials (CONSORT) Statement for nonpharmacological studies [17].

The study biochemical data are shown in in Table 1. No significant changes in serum albumin were observed in this study. Compared to baseline, serum creatinine increased, and eGFR decreased significantly at the 3rd and 6th months. Diabetic patients showed insignificant changes in eGFR; however, there was an increasing trend in eGFR in the last half of the study. For non-diabetic patients, eGFR decreased significantly at the 6th month. Additionally, HbA1c and HOMA-IR increased significantly, while LDL decreased significantly at the 6th month. Other metabolic parameters did not significantly differ at either the 3rd or the 6th month.

The nutritional and anthropometric profiles are reported in Table 2. Daily energy intakes were significantly elevated at the 3rd and 6th months (+183.35 ± 209.47 kcal/day, *p* < 0.0001 at 3rd month; +188.98 ± 225.21 kcal/day, *p* < 0.0001 at the 6th month). Protein intakes showed an insignificant change compared to baseline. The mean body weight and BMI were significantly increased at the 3rd and 6th months. Total body mineral and total body fat mass were significantly elevated at the 6th month. Handgrip strength was significantly improved at both time points, while the QoL scores (total or domain-related) were maintained (data not shown).

Table 3 describes the changes in the inter-group differences of glycemia-related data stratified by diabetes status. The changes of the glycemic-related data were not significantly different between diabetic and non-diabetic patients at the 3rd or 6th month, except for fasting blood glucose. Changes in fasting blood glucose at the 6th month were significantly higher for diabetic patients.

## 4. Discussion

In this study, we tested whether a RONS in addition to an LPD prescription for patients with CKD at stage 3b to 5 could maintained their nutritional status and quality of life, using a before–after design. Although the case number was modest, our pilot study showed that incorporating a RONS to the regimen of an LPD significantly improved energy intake and maintained the nutritional status of patients with stage 3b–5 CKD. We observed no change in serum albumin, which was a positive finding, and significant increases in body weight and handgrip strength among the study participants, suggesting that our approach of dietary supplementation might preserve or improve CKD patients’ nutritional status. These findings lend support to the notion that a nutritional intervention may benefit CKD patients requiring dialysis and even those not requiring it and that such approach potentially enhances CKD patients’ quality of life.

Findings from prior meta-analyses identified that oral nutrition supplements (ONS) might improve the nutritional status and potentially reduce complications in patients under chronic dialysis [10,18]. RONS, with a specific focus on reducing sodium, phosphorus, and potassium content relative to conventional ONS, have also been shown to restore dialysis patients’ serum albumin, increase their dry weight, and potentially lower inflammation [19]. The correction of malnutrition is presumed to attenuate dialysis patients’ risk of protein–energy wasting (PEW), potentially prolonging their survival and decreasing the healthcare burden [20]. Indeed, a recent Cochrane meta-analysis concluded that ONS increased dialysis patients’ serum pre-albumin, albumin, and anthropometric parameters, especially in those who were malnourished, although a mortality benefit was inconsistent [21]. On the other hand, among non-dialysis patients with CKD, evidence regarding the effect of combining LPD with renal-specific ONS on their outcomes remains limited [15,22,23]. A prior study recruiting 33 patients with variable CKD severities receiving an in-house RONS for replacing inter-meal snacks found patients’ weight increased significantly, with unaltered serum albumin and renal function [23]. Another short-term one-week intervention study reported that the administration of one type of RONS for replacing 30% daily energy improved energy intake without disturbing electrolyte/acid-base balances or compromising renal function among patients with stages 3 to 4 non-diabetic CKD [22]. Montes Delgado et al. found that 6 months of RONS use in patients with CKD already receiving LPD improved their adherence to the LPD prescription, better maintained patients’ nutritional status, and preserved their renal function compared to those not receiving RONS [15]. It is worth noteing that RONS use in patients with non-dialysis CKD potentially brings advantages similar to those in patients under chronic dialysis, including albumin restoration and nutritional parameter maintenance. Moreover, the above findings further suggested that RONS might obviate barriers to the continuous implementation of LPD through complication reduction and sharpening adherence. Our findings further add to the current evidence by showing that RONS may enhance muscular performance (better hand grip strength) and decrease body fat without causing electrolyte imbalances or increasing inflammation severity among non-dialysis patients with CKD (Table 1 and Table 2). As sarcopenia and frailty have been found to coexist with poor appetite and taste dysfunction in patients with CKD [24,25], it is expected that the administration of RONS to those with non-dialysis CKD and sarcopenia/frailty may prevent their further nutritional worsening and potentially improve outcomes.

In patients with advanced CKD, achieving optimal energy intake is also an important task. In this study, registered dieticians recommended a daily 1695 kcal and 45 g protein intake to the participants, and the actual energy intake increased from 1471 to 1660 kcal/day under the dieticians’ supervision. Protein intake presumably did not increase significantly during the study period compared to that at baseline, as reflected by the stable serum albumin level. In addition, participants’ energy intake conformed to the Kidney Disease Outcomes Quality Initiative (KDOQI) guideline recommendations [5]. However, at the end of this study (6th month), we observed slightly decreased eGFR levels compared to the baseline values. Possible explanations for this phenomenon include dietary changes, an increased skeletal muscle mass among the participants, and/or the progression of CKD severity. After further analyses, we discovered that the rate of eGFR decline was lower between the 3rd and the 6th month (26.40 to 26.11 mL/min/1.73 m^2^) than between baseline and the 3rd month (27.77 to 26.40 mL/min/1.73 m^2^). In the Modification of Diet in Renal Disease (MDRD) study, the average rate of decline of GFR was 3.8 mL/min/year among patients with baseline GFRs from 25 to 55 mL/min/1.73 m^2^ [26]. Kalantar-Zadeh and Foque previously revealed that a reduction in protein intake could reduce afferent arteriole perfusion pressure and intraglomerular pressure, limiting the reduction in GFR; meanwhile, the amelioration of interstitial fibrosis could further lessen kidney damage [27]. In this study, serum creatinine increased slightly by 0.10 ± 0.28 mg/dL from baseline to the 3rd month and by 0.07 ± 0.31 mg/dL from the 3rd month to the 6th month, among the 35 patients with stages 3b–5 CKD. We believe that it is more likely the different participants’ CKD stages and the increase in skeletal muscle mass, although not statistically significant, contributed to the observed elevation of creatinine level. Moreover, an initial reduction in eGFR may be translated into long-term renal function benefits, according to long-term results involving eGFR trajectories among CKD patients receiving renin–angiotensin system blockers [28].

Our results also showed that the handgrip strength significantly increased at both the 3rd and the 6th month and the values were higher than the cutoff value defined by the Asian Working Group for Sarcopenia (muscle mass < 26 kg in men and <18 kg in women) [29]. Handgrip strength, a prognostic marker in Taiwanese patients with non-dialysis CKD [30], also increased significantly during our study. These data suggest that prescribing an LPD with RONS can assist in satisfying the nutritional requirement among patients with non-dialysis CKD by providing additional calories and nutrients and preventing protein catabolism, followed by less muscle mass loss.

We found a significant increase in HOMA-IR and HbA1c at the 6th month compared to baseline, although the glucose or insulin levels did not significantly increase. Differences in changes involving glucose, but not HOMA-IR, from baseline, were significant between patients with and without diabetes mellitus. Insulin resistance is common in patients with CKD and is closely linked to increased inflammation [31]. Insulin levels frequently fluctuate in patients with CKD [32]. Nonetheless, fluctuations in insulin levels among our study participants might not be significant enough to influence their clinical outcomes. Besides, mean HbA1c significantly increased at the 6th month among all participants including the non-diabetic ones. Although it is possible that the presence of CKD increased participants’ HbA1c levels slightly [31,33], the minor changes in HbA1c levels in this study and their interpretation may not be clinically important, since multiple factors may affect the results of the HbA1c assay [34]. Nonetheless, how to optimally manage blood glucose in patients with CKD is a challenging issue, and the quality of dietary carbohydrate remains a critical element in nutritional education and related interventions. Since we used a RONS containing carbohydrates with a low glycemic index in this study, we believe that RONS might not interfere with glycemic control in patients with non-dialysis CKD.

Interestingly, we showed that the administration of RONS to patients with non-dialysis CKD rendered their QoL stationary. Frequently, QoL decreases as the severity of CKD rises [35,36], albeit study results can be heterogenous [37]. These discrepancies in reported findings may stem from the tools used, the etiology of CKD, the degree of patient’s adherence to their prescriptions of RONS, etc. In this study, we suggest that our dieticians might have exerted a positive influence on the patients through educating them and their caregivers about the benefits of LPD as well as of RONS. Such indirect influence might increase patients’ adherence to a trial regimen and enhance the effect of RONS.

Our study has its strengths and limitations. A major limitation of this study is its lack of a control group and the relatively small sample size for a single-arm study. However, given the meager evidence addressing the utility of RONS use in patients with non-dialysis stages 3b–5 CKD, our pilot and proof-of-concept study is expected to partly answer the question whether RONS influence nutritional endpoints and quality of life in these patients. For patients with CKD already receiving multidisciplinary care including a regular dietician input, the addition of RONS to the existing nutritional care regimen helps them meet their goals of adequate energy and protein intake. Potential benefits include a healthier body weight, greater muscle mass, and greater handgrip strength, while maintaining nutritional status and serum albumin levels. On the contrary, such measure should be closely monitored, in order to detect occult progression of CKD severity. A comprehensive CKD care strategy must be undertaken to reduce the socio-economic burdens patients already sustain. Based on our findings, we suggest that RONS may serve as an alternative dietary supplementation option for patients with advanced CKD. A controlled, large-scale, prospective study including patients with early or non-dialysis CKD may be needed to validate our results and to determine whether an LPD with RONS truly benefits patients with an earlier stage of CKD.

## 5. Conclusions

In conclusion, our results support the conclusion that patients with non-dialysis advanced CKD (stages 3b–5) already receiving multi-disciplinary pre-dialysis care may benefit from additional RONS besides their regular diet. Assisted by dieticians for promoting patients’ adherence to LPD, patients with advanced CKD receiving RONS might achieve better nutrition without accelerated renal function decline. Larger trials including patients with earlier CKD are required for verification.

## Figures and Tables

**Figure 1 jpm-11-01360-f001:**
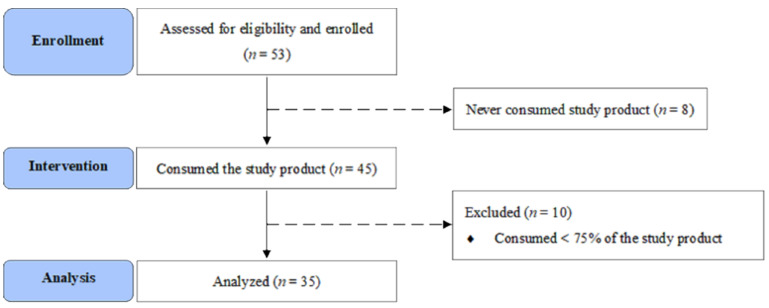
CONSORT flow diagram.

**Table 1 jpm-11-01360-t001:** Biochemical parameters at the three study time points for patients with CKD Stage 3b–5 receiving a renal-specific oral nutrition supplement (RONS).

	Baseline (*n* = 35)	3rd Month (*n* = 35)	*p*-Value ^‡^	6th Month (*n* = 35)	*p*-Value ^§^
Blood chemistry					
Albumin (g/dL)	4.13 ± 0.31 (4.10)	4.12 ± 0.32 (4.10)	0.429	4.12 ± 0.32 (4.10)	0.407
Blood urea nitrogen (mg/dL)	34.49 ± 12.58 (32.00)	35.83 ± 11.80 (37.00)	0.245	34.94 ± 12.67 (34.00)	0.749
Creatinine (mg/dL)	2.34 ± 0.76 (2.20)	2.45 ± 0.81 (2.30)	0.037 *	2.51 ± 0.96 (2.30)	0.016 *^,^^†^
eGFR (ml/min/1.73 m^2^)	27.77 ± 10.02 (28.00)	26.40 ± 9.54 (25.00)	0.044 *	26.11 ± 9.97 (26.00)	0.031 *
Nondiabetic group (*n* = 26)	25.69 ± 10.12 (25.00)	24.50 ± 9.73 (21.50)	0.161	24.00 ± 9.84 (23.00)	0.042 *
Diabetic group (*n* = 5)	33.78 ± 7.24 (33.00)	31.89 ± 6.75 (32.00)	0.092	32.22 ± 7.98 (32.00)	0.403
Uric acid (mg/dL)	6.52 ± 1.62 (6.20)	6.39 ± 1.56 (6.30)	0.277	6.50 ± 1.61 (6.30)	0.931
HbA1c (%)	5.77 ± 0.75 (5.60)	5.87 ± 0.70 (5.70)	0.079	5.91 ± 0.78 (5.70)	0.021 *^,^^†^
Glucose (mg/dL)	103.46 ± 25.06 (99.00)	100.54 ± 12.76 (102.00)	0.627 ^†^	103.54 ± 24.68 (103.00)	0.107 ^†^
Insulin (µIU/mL)	6.55 ± 4.61 (5.40)	8.77 ± 10.20 (5.70)	0.311 ^†^	8.63 ± 10.83 (6.00)	0.119 ^†^
HOMA-IR	1.74 ± 1.46 (1.25)	2.05 ± 2.32 (1.35)	0.563 ^†^	2.51 ± 4.49 (1.61)	0.023 *^,^^†^
Cholesterol (mg/dL)	190.74 ± 36.20 (191.00)	180.43 ± 40.93 (173.00)	0.187	180.77 ± 48.69 (176.00)	0.203
Triglycerides (mg/dL)	124.69 ± 55.65 (116.00)	136.00 ± 77.36 (110.00)	0.182	146.43 ± 99.64 (133.00)	0.098 ^†^
LDL (mg/dL)	108.74 ± 28.46 (109.00)	98.09 ± 30.68 (88.00)	0.072	95.57 ± 36.37 (89.00)	0.026 *
Sodium (mmol/L)	140.77 ± 2.56 (141.00)	140.54 ± 2.31 (141.00)	0.481	140.29 ± 2.70 (141.00)	0.155
Potassium (mmol/L)	4.38 ± 0.58 (4.30)	4.43 ± 0.49 (4.40)	0.382	4.42 ± 0.65 (4.40)	0.583
Calcium (mg/dL)	9.26 ± 0.33 (9.20)	9.21 ± 0.38 (9.20)	0.362	9.18 ± 0.41 (9.20)	0.205
Phosphorus (mg/dL)	3.85 ± 0.67 (3.70)	3.84 ± 0.68 (3.70)	0.943	3.95 ± 0.67 (4.00)	0.148
CRP (mg/L)	1.13 ± 1.43 (0.50)	2.22 ± 4.23 (0.50)	0.099	2.12 ± 4.09 (0.50)	0.166 ^†^

eGFR, estimated glomerular filtration rate; HbA1c, glycated hemoglobin; HOMA-IR, Homeostatic Model Assessment for Insulin Resistance; LDL, low-density lipoprotein; CRP, C-reactive protein. Data are presented as mean ± SD (median). ^†^ Signed-Rank Test was used after the Shapiro–Wilk test as the data did not follow a normal distribution. ^‡^ *p*-value of change from baseline to the 3rd month. ^§^ *p*-value of change from baseline to the 6th month. * Statistical significance (*p* < 0.05).

**Table 2 jpm-11-01360-t002:** Nutritional and anthropometric parameters of the for patients with CKD Stage 3b–5 receiving a renal-specific oral nutrition supplement (RONS).

	Baseline (*n* = 35)	3rd Month (*n* = 35)	*p*-Value ^‡^	6th Month (*n* = 35)	*p*-Value ^§^
Nutritional status and intakes					
Body weight (kg)	61.67 ± 11.35 (63.30)	62.76 ± 10.93 (64.75)	< 0.001 *	62.98 ± 10.85 (65.00)	< 0.001 *
Body Mass Index (kg/m^2^)	23.58 ± 3.43 (24.02)	24.01 ± 3.27 (24.84)	< 0.001 *	24.10 ± 3.26 (24.55)	< 0.001 *
Daily energy intake (kcal/d)	1470.77 ± 330.69 (1486.67)	1654.12 ± 251.59 (1630.00)	< 0.001 *	1659.75 ± 245.40 (1595.67)	< 0.001 *
Total energy intake (kcal/kg/d)	24.14 ± 5.11 (24.35)	26.81 ± 4.68 (26.78)	< 0.001 *	26.82 ± 4.79 (25.98)	< 0.001 *
Total protein intake (g/d)	47.98 ± 12.81 (48.33)	49.90 ± 11.64 (48.10)	0.277	50.79 ± 11.63 (48.60)	0.140
Total protein intake (g/kg/d)	0.78 ± 0.19 (0.81)	0.80 ± 0.16 (0.80)	0.612	0.81 ± 0.18 (0.77)	0.358
Body composition and handgrip strength					
Total body protein (kg)	8.67 ± 1.54 (8.10)	8.76 ± 2.01 (8.50)	0.801 ^†^	8.76 ± 1.76 (8.55)	0.379 ^†^
Total body mineral (kg)	2.95 ± 0.51 (2.88)	2.98 ± 0.60 (2.87)	0.230 ^†^	3.02 ± 0.52 (2.98)	0.005 *
Skeletal muscle mass (kg)	24.12 ± 4.65 (22.50)	24.41 ± 6.02 (23.70)	0.586 ^†^	24.99 ± 6.33 (23.80)	0.216 ^†^
Fat-free mass (%)	62.19 ± 15.78 (64.30)	60.24 ± 17.10 (61.60)	0.682 ^†^	58.99 ± 17.64 (61.85)	0.854 ^†^
Total body fat mass (kg)	16.93 ± 7.16 (17.70)	17.58 ± 6.84 (17.10)	0.070 ^†^	18.09 ± 6.56 (18.00)	0.011 *^,^^†^
Body fat (%)	27.47 ± 10.77 (28.10)	28.71 ± 12.59 (26.60)	0.175 ^†^	28.39 ± 11.10 (26.65)	0.707 ^†^
Handgrip strength (kg)	26.97 ± 7.71 (27.50)	27.54 ± 7.54 (27.32)	0.038 *	28.00 ± 7.87 (28.50)	0.036 *^,^^†^

eGFR, estimated glomerular filtration rate. Data are presented as mean ± SD (median). ^†^ Signed-Rank Test was used after the Shapiro–Wilk test as the data did not follow a normal distribution. ^‡^ *p*-value of change from baseline to the 3rd month. ^§^ *p*-value of change from baseline to the 6th month. * Statistical significance (*p* < 0.05).

**Table 3 jpm-11-01360-t003:** Glycemia-related data at each study time point for the nondiabetic group compared to the diabetic group.

	Baseline		3rd Month		*p*-Value ^‡^	6th Month		*p*-Value ^§^	*p*-Value ^¶^
	Nondiabetic (*n* = 26)	Diabetic (*n* = 9)	Nondiabetic (*n* = 26)	Diabetic (*n* = 9)		Nondiabetic (*n* = 26)	Diabetic (*n* = 9)		
Glucose (mg/dL)	99.42 ± 8.86 (99.50)	115.11 ± 47.10 (96.00)	99.27 ± 10.50 (101.50)	104.22 ± 18.07 (110.00)	0.408	96.46 ± 20.96 (102.00)	124.00 ± 24.14 (120.00)	0.820	0.012 *
HOMA-IR	1.50 ± 1.13 (1.19)	2.44 ± 2.10 (1.64)	1.73 ± 1.40 (1.37)	2.95 ± 3.89 (1.28)	0.549	1.69 ± 1.23 (1.48)	4.90 ± 8.49 (1.76)	0.128 ^†^	0.092
HbA1c (%)	5.48 ± 0.23 (5.50)	6.62 ± 1.08 (6.40)	5.60 ± 0.30 (5.60)	6.63 ± 0.94 (6.60)	0.408	5.63 ± 0.42 (5.55)	6.73 ± 1.02 (6.60)	0.649	0.985 ^†^
Insulin (µIU/mL)	5.93 ± 4.10 (5.05)	8.34 ± 5.73 (6.90)	7.45 ± 5.77 (5.85)	12.59 ± 17.78 (4.70)	0.162	6.89 ± 4.49 (6.00)	13.67 ± 19.92 (6.00)	0.327	0.365

Data are presented as mean ± SD (median). The analysis of covariance model was fitted for the variables’ change from baseline to the 3rd month and from baseline to the 6th month for comparison between the diabetic and the non-diabetic groups with baseline as a covariate. For change from the 3rd month to 6th month, the analysis of variance model was fitted. When the data were not normal, the two-sample two-sided Wilcoxon rank sum test was used to compare the groups. ^†^ Signed-Rank Test was used after the Shapiro–Wilk test as the data did not follow a normal distribution ^‡^ Difference in changes of values between the nondiabetic and the diabetic groups at baseline versus the 3rd month. ^§^ Difference in changes of values between the nondiabetic and the diabetic groups at the 3rd month versus baseline. ^¶^ Difference in changes of values between the nondiabetic and diabetes groups at the 6th month versus baseline. * Statistical significance (*p* < 0.05).

## Data Availability

The data shown in this study are available on request from the corresponding author.

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
