# Peer review of "A Low-Protein Diet with a Renal-Specific Oral Nutrition Supplement Helps Maintain Nutritional Status in Patients with Advanced Chronic Kidney Disease"

_jpm, 2021, doi:10.3390/jpm11121360_

Round 1

Reviewer 1 Report

1, The number of inclusion in the appropriate disease population in the manuscript as well as the number of inclusion in controls are not clearly explained, and the formula for the sample size estimate should be given.

2, In the "Sample Size Calculation" paragraph, what does the author describe?

3, In figure1, I saw only 35 patients included in the study. Is the amount of data too small? This is not a rare disease, after all.

4, In methodology, the authors should directly illustrate the meaning of midpoint, end of study, and Baseline, respectively, rather than in the table.

Author Response

Please refer to the attachment titled as response letter. Thank you.

Reviewer 2 Report

This is a straight forward clinical study to show that specific RONS nutrient management are effective to maintain nutritional status and quality of life, in stage 3b to 5 CKD patients. Such studies provide useful information for the caring of this specific type of patients by using respective diet.

The rationale is based on previous prior meta-analyses that identified ONS
might improve nutritional status and reduce complications in patients under
chronic dialysis. The studies are well planned and designed and results are well analyzed.

Despite the moderate number of enrollment, I feel the results are significant and statistics are convincing.

The writing is in a simple but clear way. It is very easy to follow. One critique: as there are so many abbreviations, it might be helpful it authors could provide abbreviation list.

Author Response

Thanks for your comments. We have provided an abbreviation list in the manuscript.

Round 2

Reviewer 1 Report

Thanks to the authors for the revision, the manuscript has been promoted.